# Innovative behavior and organizational innovation climate among the Chinese clinical first-line nurses during the Omicron pandemic: The mediating roles of self-transcendence

**Zhangyi Wang**[1], **Yue Zhu**[2], **Tingrui Wang**[3], **Tao Su**[2], **Huifang Zhou**[1], **Siai Zhang**[4], **Mengru Liu**[5], **Lamei Chen**[1], **Manli Wu**[6], **Liping Li**[1], **Xuechun Li**[7], **Xiaoli Pang**[2], **Jiaofeng Peng**[1], **Xiaochun Tang**[1], **Li Liu**[8]*

1 Nursing Department, Affiliated Hengyang Hospital of Hunan Normal University & Hengyang Central Hospital, Hengyang, Hunan Province, China, 2 School of Nursing, Tianjin University of Traditional Chinese Medicine, Tianjin, China, 3 School of Nursing, Guizhou Medical University, Guiyang, Guizhou Province, China, 4 Cardiac Intensive Care Unit (CICU), Meizhou People's Hospital (Meizhou Academy of Medical Sciences), Meizhou, Guangdong Province, China, 5 Kidney Transplantation Department, Shandong Provincial Hospital Affiliated to Shandong First Medical University, Jinan, Shandong Province, China, 6 School of Nursing, Gansu University of Chinese Medicine, Lanzhou, Gansu Province, China, 7 School of Nursing, Dali University, Dali, Yunnan Province, China, 8 Nephrology Department / Rheumatology and Immunology Department, The Second Affiliated Hospital of University of South China, Hengyang, Hunan Province, China

* 170243406@qq.com

**Data Availability Statement:** All relevant data are within the paper and its Supporting information files.

## Abstract

### Background

During the Omicron pandemic, clinical first-line nurses played a crucial role in healthcare. Their innovative behavior enhanced the quality of nursing and served as a vital factor in driving the sustainable development of the nursing discipline and healthcare industry. Many previous studies have confirmed the significance of nurses' innovative behavior worldwide. However, the correlations among innovative behaviors, organizational innovation climate, self-transcendence, and their mediating roles in Chinese clinical first-line nurses need further research.

### Methods

A cross-sectional study was conducted, and the quality reporting conformed to the STROBE Checklist. From March 2022 to February 2023, a convenience sample of 1,058 Chinese clinical first-line nurses was recruited from seven tertiary grade-A hospitals of Tianjin city in Northern China. The Demographic Characteristics Questionnaire, Nurse Innovative Behavior Scale (NIBS), Nurse Organizational Innovation Climate Scale, and the Self-Transcendence Scale were used. The data was analyzed using descriptive statistics, correlation, and process plug-in mediation effect analyses.

**Funding:** This study was supported by the Tianjin Research Innovation Project for Postgraduate Students (CN) [grant numbers: 2021YJSS171], the Scientific Research Project of Hunan Provincial Health Commission (CN) [grant numbers: D202314017358], the Health Research Project of Hunan Provincial Health Commission (CN) [grant numbers: W20243278], the Hengyang Science and Technology Plan Project (CN) [grant numbers 202222035776], and the Tianjin University of Traditional Chinese Medicine Science and Technology Innovation Fund Project for College Students (CN) [grant numbers: ZX01]. The funders of Zhangyi Wang, Li Liu, Xiaochun Tang, Yue Zhu, and Liping Li had a role in study design, data collection and analysis, decision to publish, and preparation of the manuscript.

**Competing interests:** The authors have declared that no competing interests exist.

## Results

The total scores of innovative behavior, organizational innovation climate, and self-transcendence were 33.19 ± 6.71, 68.88 ± 12.76, and 41.25 ± 7.83, respectively. Innovative behavior was positively correlated with the organizational innovation climate ($r = 0.583$, $p < 0.01$) and self-transcendence ($r = 0.635$, $p < 0.01$). Self-transcendence partially mediated mediating role between innovative behavior and organizational innovation climate, accounting for 41.7%.

## Conclusion

The innovative behavior, organizational innovation climate, and self-transcendence among the first-line nurses during the Omicron pandemic were relatively moderate, which needs improving. Organizational innovation climate can directly affect the innovative behavior among Chinese clinical first-line nurses and indirectly through the mediating role of self-transcendence. It is recommended that nursing managers adjust their management strategies and techniques based on the unique characteristics of nurses during the pandemic. This includes fostering a positive and inclusive environment for organizational innovation, nurturing nurses' motivation and awareness for innovation, enhancing their ability to gather information effectively, overcoming negative emotions resulting from the pandemic, and promoting personal growth. These efforts will ultimately enhance nursing quality and satisfaction during the Omicron pandemic.

## 1. Introduction

Coronavirus Disease 2019 (COVID-19) is the largest atypical pneumonia outbreak since the Severe Acute Respiratory Syndrome (SARS) in 2003 [1]. Omicron refers to the mutant strain of COVID-19. Compared to Delta or SARS-CoV2, Omicron has more vital transmission ability, faster transmission speed, shorter incubation period, and rapid disease progression. These characteristics have had a profound negative influence on public health and have significantly impacted patients and clinical first-line nurses [2, 3]. By the end of 2022, many nurses in China were deployed to the frontlines of clinical care to assist in managing and containing the Omicron pandemic [4]. However, clinical first-line nurses faced a greater risk of contracting Omicron than the general population [1]. Not only do they have to endure the demanding workload brought about by the pandemic, but they also need to continuously innovate traditional nursing models and methods in clinical practice and enhance innovative awareness and ability to adapt to the pandemic-induced changes [5–7]. And many studies have demonstrated that clinical first-line nurses were known to be at risk for depression, anxiety, fear, post-traumatic stress disorder, and enthusiasm for innovation to varying degrees during the Omicron pandemic, significantly higher than the SARS pandemic or general population [8, 9]. Studies have shown that, improving nurses' innovative behavior is a robust measure for addressing resource scarcity, heavy clinical nursing burden, and managing the Omicron pandemic challenges [10]. At the same time, the innovation of nurses is receiving greater global attention. During the Omicron pandemic, the World Health Organization (WHO) suggested that some new treatments and nursing techniques that benefit the prognosis of patients should be innovated by improving the innovative behavior of clinical first-line nurses [11]. And in China, the

National Health Planning Commission also clarified the importance of innovation in the medical and health fields, emphasized the need to enhance the innovation ability of the medical staff, and continuously innovate the nursing service model by strengthening the scientific and technological innovation systems [12]. However, few research were on the innovative behavior of clinical first-line nurses and its related correlation study during the Omicron pandemic in China. Therefore, it is essential to investigate the innovative behavior of clinical first-line nurses and explore the relationships between other related variables.

## 1.1. Literature review

Many foreign scholars defined innovative behavior as intentional generation, promotion, and realization of new ideas within a work role, group, or organization, which can determine the dynamic development of nursing career [13–15]. In addition, Bao's [16] definition of nurses' innovative behavior is widely accepted in China. According to this definition, nurses seek and develop new methods, technologies, and working methods to promote health, prevent diseases, and enhance the quality of patient care. Ample evidence indicates that many factors, including individual and environmental factors, influence the innovative behavior of nurses [17]. Information literacy, psychological resilience, and psychological empowerment are the individual factors that influence nurses' innovative behavior [18, 19], and innovation climate perception, authentic leadership, and organizational support are the environmental factors related to the innovative behavior of nurses [20].

**1.1.1. Organizational innovation climate can positively predict innovative behavior.** Organizational innovation climate refers to the external environment and conditions that drive groups or individuals to develop innovative thinking and behavior. It also includes nurses' consistent cognitive experience of innovative characteristics of the working environment in hospitals and departments. In essence, it is the perception of individuals for innovative orientation, characteristics, and support of the internal organization environment [21]. Organizational innovation climate can provide a cognitive basis for innovation and support the formation of innovative behavior [22]. Studies have shown that an excellent organizational innovation atmosphere is conducive to improving work enthusiasm and innovation of nurses and can positively predict the occurrence of their innovative behavior [23, 24]. Another study has indicated that organizational innovation climate stimulated the innovative thinking of college students and cultivated their innovative behavior to a certain extent, proving that the organizational innovation climate positively influenced the individual's innovative behavior [21]. The organizational innovation climate is vital for clinical first-line nurses in improving their innovation behavior. An effective organizational innovation climate can provide nurses with resources, psychological support, and the latest technology, which can facilitate the generation of more innovative behavior [22].

**1.1.2. Self-transcendence is a positive predictor of innovative behavior.** Self-transcendence is an internal and specific psychological experience that refers to expanding the boundaries of personal ability in many ways and constantly surpassing the status quo to achieve a higher level of self-realization [25]. Self-transcendence is also a process of spiritual development in adulthood, involving the pursuit of harmony with nature and a sense of identity with others or oneself [26, 27]. A high self-transcendence enables individuals to adjust their cognitive and behavior patterns, respond to life events with a positive attitude, and grow from their setbacks [28]. Once individuals transcend themselves, they will attain a positive life, confidence, and maturity and can cope better with physical, mental, and emotional stress. Studies have confirmed that nurses with a high level of self-transcendence are more energetic and

focused in their work and also have a stronger sense of job involvement and higher innovation ability and behavior [29, 30].

**1.1.3. The theoretical framework of this study was Space Theory [31, 32], proposed according to the field dynamics theory [33].** According to the theory of field dynamics, the formula of innovation behavior is $Bi = f \cdot (Si)$, where innovation behavior ($Bi$) is a function of the innovation space, $Si$. According to this theory, innovation, as a social behavior, also has an innovative situation, and this situation is the innovative space. Therefore, another form of innovative behavior formula is finally obtained: $Bi = f (Pi \cdot Ei)$, where $Bi$ stands for behavior (including innovative behavior), $Pi$ stands for the person (including personal internal needs, internal psychological factors, and so forth), and $Ei$ stands for the environment (including innovative environment and external environmental factors). Furthermore, it is important to note that the innovation subject $Pi$ and the innovation environment $Ei$ are interdependent variables that comprise $Si$. $Ei$ can be understood as the sum of social and natural relations affecting $Bi$ and $Si$, which can be regarded as the dynamic field of innovation behavior. In $Si$, $Pi$ and $Ei$ will interact. Based on the theories of space and field dynamics, the innovative behavior of nurses can be explained as a response to external factors that disrupt their usual working state. Nurses engage in innovative behavior to adapt to their environment, restore balance, and continue their work in nursing. During the Omicron pandemic, clinical first-line nurses must ensure the quality of care while facing greater work pressure, which may give birth to innovative behavior of nurses. Furthermore, this demonstrates how external factors impact the internal factors of nurses and promote their innovative behavior. In particular, combined internal and external factors can affect the innovative behavior of nurses, and may affect the innovative behavior of nurses through the intermediary role of internal factors.

In conclusion, the correlations among innovative behaviors, organizational innovation climate, self-transcendence, and their mediating roles in Chinese clinical first-line nurses is still open for further research. Therefore, based on the space theory, internal personal (self-transcendence) and external environmental factors (organizational innovation climate) will restrict and promote each other and eventually work together on the innovative behavior of clinical first-line nurses. Thus, based on this theory and literature review, the conceptual framework of this study was constructed, as shown in Fig 1.

## 2. Objectives

The aims of this study are (1) to investigate the innovative behavior, organizational innovation climate, and self-transcendence among the Chinese clinical first-line nurses, (2) to examine the correlations among innovative behavior, organizational innovation climate, and self-transcendence, (3) to explore the mediating role of self-transcendence between innovative behavior and organizational innovation climate, and (4) to provide a theoretical basis for constructing the interventional measures to improve the innovative behavior and ability of the Chinese clinical first-line nurses, further improving the nursing quality and satisfaction during the Omicron pandemic, ultimately developing clinical nursing.

## 3. Methods

### 3.1. Study design and setting

This study used a descriptive cross-sectional design and was conducted of Tianjin city in Northern China from March 2022 to February 2023. The quality reporting of the study adhered to the STROBE Checklist (see S1 File).

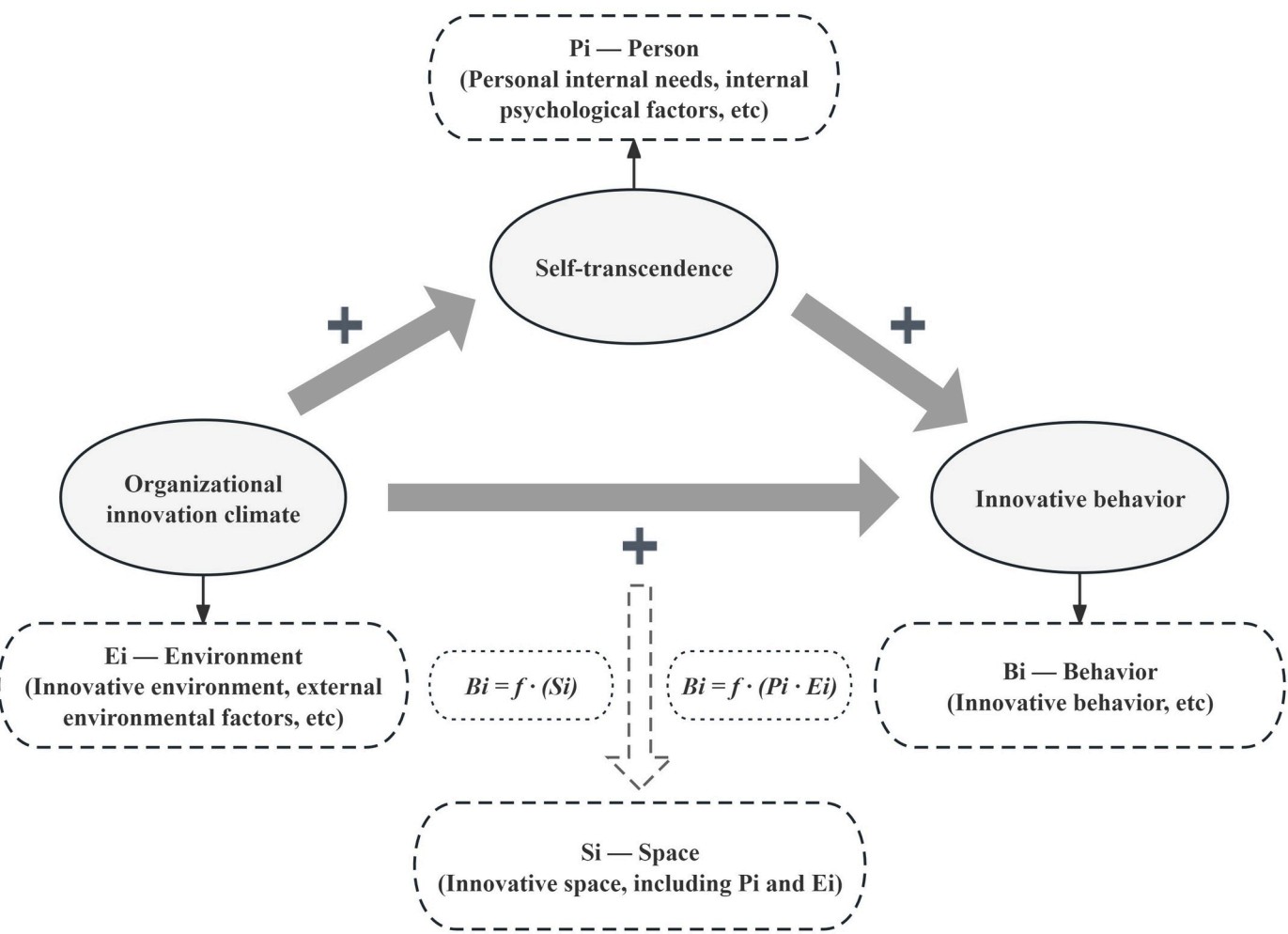

**Fig 1. The conceptual framework of innovative behavior, organizational innovation climate, and self-transcendence of the study.**

## 3.2. Participants and sample

**3.2.1. Participants.** Convenience sampling was used to recruit the nurses from seven tertiary grade-A hospitals during the Omicron pandemic, which are all located in Tianjin city of Northern China, with more than 1,500 beds and 2,000 employees. Respondents met the following criteria—inclusion criteria: (1) obtained a nurse's professional qualification and register certificates; (2) Worked in the first-line during the Omicron pandemic for $\geq 1$ month. Exclusion criteria: (1) intern nurses, (2) advanced training, rotation, and regular training nurses, and (3) nurses who were not on duty during the investigation period. Written informed consents were taken from the nurses who volunteered to participate in the study.

**3.2.2. Sampling methods and sample size calculation.** Participants were recruited from seven tertiary grade-A hospitals of Tianjin city in Northern China from March 2022 to February 2023. The investigation was conducted with the hospitals' prior approval, focusing on individual departments and with the assistance of the head nurse in each department for distribution. We used the unified instruction language to explain the essential information to the participants, including the purpose, significance, and confidentiality of this study, and the participants were investigated face-to-face using an online questionnaire. The questionnaires

underwent quality control through several measures: (1) each micro-signal can be filled only once, after being authorized by the system settings; (2) the participants are required to answer all the questions; (3) questionnaires with completion time less than one minute were excluded; (4) questionnaires with obvious contradictory answers were excluded; (5) questionnaires with obvious inconsistent personal information of the respondents were excluded. Finally, 1,058 valid questionnaires were collected, and the effective recovery rate was 96.7%.

G*Power 3.1.9.7 software [34] was used to calculate the minimum sample size required for $F$-tests [α error probability was 0.05, and power (1—β error probability) was 95%]. Because all variables involved of this study are continuous, therefore, the $F$-tests method was selected in G*Power 3.1.9.7 software, "Linear multiple regression: Fixed model, $R^2$ deviation from zero" of statistical test was selected, "A priori: Compute required sample size—given α, power, and effect size" of type of power analysis was selected, and then the parameter effect size $f^2 = 0.15$, α = 0.05, 1—β = 0.95 were set, and finally clicked "Calculate" to calculate the total sample size = 107, the minimum sample size of this study.

Considering the hospital and clinical realistic environment, 1,058 participants were included in this study in the end. And "Post hoc: Compute achieved power—given α, sample size, and effect size" of type of power analysis was selected. And then the parameter effect size $f^2 = 0.15$, α = 0.05, total sample size = 1058 were set, and finally clicked "Calculate" to calculate the 1—β = 1.00, which was greater than the original 1—β of 0.95 and indicating that the sample size of the study meets the sample size required for the mediation analysis. In addition, participants were recruited from seven tertiary grade-A hospitals of Tianjin city in Northern China through online questionnaires, and collected more questionnaires to ensure the generalization and universality of the research results, so as to minimize the deviation of the results caused by geography and hospital nature. In this way, included 1,058 participants were necessary and would greatly enhance the transparency and understanding of the study design, especially in relation to the mediation analysis.

### 3.3. Measurements

**3.3.1. The Demographic Characteristics Questionnaire.** The Demographic Characteristics Questionnaire was developed and used to investigate the 17 demographic characteristics of Chinese clinical first-line nurses, such as gender, age, nursing age, marital status, number of children, monthly per capita income, educational background, technical title, administrative position, nature of the hospital, employment modality, income satisfaction, and so on.

**3.3.2. The Nurse Innovative Behavior Scale.** The Nurse Innovative Behavior Scale (NIBS) was used to investigate the innovative behavior of nurses. The scale was compiled by Bao et al. [16] based on the characteristics of Chinese nurses. The scale included three dimensions and 10 items: "Generating ideas" (three items), "Obtaining support" (four items), and "Realizing ideas" (three items). The Cronbach's α and content validity of the scale were 0.879 and 0.910, respectively, which calculated in this study was 0.955. Likert five scoring method was used, with the 1–5 scores indicating "never," "less," "sometimes," "often," and "frequently," respectively. The total score ranges from 10–50 points, with higher scores indicating higher innovative behavior.

**3.3.3. The Nurse Organizational Innovation Climate Scale.** The Nurse Organizational Innovation Climate Scale (NOICS) was developed by Qian et al. [35]. The scale consists of three dimensions and 21 items, including "Organizational innovation incentives" (seven items), "Resource supply" (six items), and "Management practices" (eight items). The Cronbach's α and content validity of the scale were 0.940 and 0.938, respectively, calculated in this study was 0.979. Likert five scoring method was used, with the 1–5 scores indicating from

"strongly disagree" to "strongly agree," respectively. The higher the score, the better the organizational innovation climate.

**3.3.4. The Self-Transcendence Scale.** The Self-Transcendence Scale (STS) was created by Reed [25] and Zhang et al. [36]. It consists of one dimension and 15 items. The Cronbach's α was reported as 0.892, calculated as 0.907 in this study. The scale used a 4-point Likert type, and "non-conformance," "only some conformance," "some conformance," and "very conformance" are assigned with 1–4 points, respectively. The total score was 15–60 points, with a score greater than 45 indicating a good self-transcendence. Higher scores signified higher self-transcendence.

### 3.4. Statistical analysis

Two researchers recorded and analyzed the raw data using Epidata 3.1 and IBM SPSS 21.0. Descriptive statistics (numbers and percentage distribution) were used to describe the demographic characteristics. Mean ± Standard deviation [M (SD)] described measurement data meeting the normal distribution. Pearson correlation analysis was used to explore their correlations if the variables conform to normal distribution. Otherwise, Spearman correlation analysis was used. PROCESS plug-in mediation effect analysis [37] was used to investigate the mediating role of self-transcendence between innovative behavior and organizational innovation climate with the statistical significance set to $p < 0.05$ (two-tailed).

### 3.5. Ethics approval and consent to participate

The study protocol was approved by the Medical Ethics Committee of Tianjin University of Traditional Chinese Medicine (Approval number: 2022-027-18). Written informed consents were obtained from all the study participants. All the procedures were performed according to the principles of the Declaration of Helsinki and following the relevant guidelines and regulations in China. After obtaining permission from hospital administrators, the researchers approached the participants with the help of the head nurses. The participants were given the right to refuse or withdraw from this study. The questionnaires were designed to maintain anonymity and confidentiality. This was achieved by ensuring no identifying marks, names, or numbers were associated with the participants. The data obtained were only used for academic research, not commercial purposes.

## 4. Results

### 4.1. Demographic characteristics

A total of 1,073 Chinese clinical first-line nurses were recruited to participate in the survey from seven tertiary grade-A hospitals located in Tianjin city of Northern China, with more than 1,500 beds and 2,000 employees. Among them, there are five general and two specialty hospitals. Besides, 15 nurses were excluded from the study because of their inconsistent answers. Hence, 1,058 nurses were eventually included in the study. Among the 1,058 nurses, 1,006 participants (95.1%) were female, with 463 subjects (43.8%) aged 26–30, with an average age of 31.85 ± 5.01 years. Five hundred twenty-five nurses (49.6%) have worked for 6–10 years. All the demographic characteristics are outlined in Table 1.

### 4.2. Scores of innovative behavior, organizational innovation climate, and self-transcendence

The total score of innovative behavior was 33.19 ± 6.71, and the average score of NIBS was 3.32 ± 0.75. Among the three dimensions of NIBS, the dimension with the highest average

**Table 1. Demographic characteristics of the Chinese clinical first-line nurses (*n* = 1,058).**

| Characteristics | *n* | % |
|---|---|---|
| **Gender** | | |
| Male | 52 | 4.9 |
| Female | 1006 | 95.1 |
| **Age (years)** | | |
| ≤ 25 | 328 | 31.0 |
| 26 ~ 30 | 463 | 43.8 |
| 31 ~ 35 | 169 | 16.0 |
| > 35 | 98 | 9.2 |
| **Nursing age (years)** | | |
| ≤ 5 | 336 | 31.8 |
| 6 ~ 10 | 525 | 49.6 |
| > 10 | 197 | 18.6 |
| **Marital status** | | |
| Single | 375 | 35.4 |
| Married | 663 | 62.7 |
| Divorced | 13 | 1.2 |
| Widowed | 7 | 0.7 |
| **Number of children** | | |
| None | 389 | 36.8 |
| 1 | 507 | 47.9 |
| ≥ 2 | 162 | 15.3 |
| **Monthly per capita income (RMB)** | | |
| < 3000 | 61 | 5.8 |
| 3000 ~ 4999 | 135 | 12.8 |
| 5000 ~ 6999 | 217 | 20.5 |
| 7000 ~ 8999 | 377 | 35.6 |
| ≥ 9000 | 268 | 25.3 |
| **Educational background,** | | |
| Technical secondary school | 8 | 0.8 |
| Junior college degree | 79 | 7.5 |
| Bachelor degree | 915 | 86.4 |
| Master degree or above | 56 | 5.3 |
| **Technical title** | | |
| Nurse | 201 | 18.9 |
| Nurse Practitioner | 534 | 50.5 |
| Nurse-in-Charge | 278 | 26.3 |
| Associate Nurse Practitioner | 38 | 3.6 |
| Chief Nurse Practitioner | 7 | 0.7 |
| **Administrative position** | | |
| None | 1021 | 96.5 |
| Head nurse | 26 | 2.5 |
| Head nurse of ward | 8 | 0.8 |
| Director of nursing department | 3 | 0.2 |
| **Nature of the hospital** | | |
| Specialized hospital | 198 | 18.7 |
| General hospital | 860 | 81.3 |
| **Whether is a specialist nurses** | | |

(*Continued*)

**Table 1.** (Continued)

| Characteristics | n | % |
|---|---|---|
| Yes | 103 | 9.7 |
| No | 955 | 90.3 |
| **Whether is a clinical instructor** | | |
| Yes | 171 | 16.2 |
| No | 887 | 83.8 |
| **Employment modality** | | |
| Enterprise system | 579 | 54.7 |
| Contractual system | 386 | 36.5 |
| Labour dispatch system | 93 | 8.8 |
| **Income satisfaction** | | |
| Dissatisfied | 166 | 15.7 |
| Average | 595 | 56.2 |
| Satisfied | 273 | 25.8 |
| Very Satisfied | 24 | 2.3 |
| **Whether have ever applied for a nursing research project** | | |
| Yes | 69 | 6.5 |
| No | 989 | 93.5 |
| **Whether have ever published a paper** | | |
| Yes | 150 | 14.2 |
| No | 908 | 85.8 |
| **Whether have attended a nursing research programme** | | |
| Yes | 87 | 8.2 |
| No | 971 | 91.8 |

Note: *: $p < 0.05$, **: $p < 0.01$

score was "Generating ideas" (3.38 ± 0.79), and the lowest was "Realizing ideas" (3.27 ± 0.68). The scores of "Generating ideas," "Obtaining support," and "Realizing ideas" were 10.14 ± 2.83, 13.24 ± 3.25, and 9.81 ± 2.72, respectively.

The total score of organizational innovation climate was 68.88 ± 12.76, and the average score of NOICS was 3.28 ± 0.83. Among the three dimensions of NOICS, the dimension with the highest average score was "Resource supply" (3.38 ± 0.91), and the lowest was "Management practices" (3.17 ± 0.83). The scores of "Organizational innovation incentives," "Resource supply," and "Management practices" were 23.24 ± 5.47, 20.28 ± 4.53, and 25.36 ± 6.26, respectively.

The total score of self-transcendence was 41.25 ± 7.83, and the average score of STS was 2.75 ± 0.68. The NIBS, NOICS, and STS scores are shown in Table 2.

## 4.3. Correlation between innovative behavior and organizational innovation climate

Pearson's correlation analysis demonstrated a significant and positive correlation between the scores of innovative behavior and organizational innovation climate ($r = 0.583$, $p < 0.01$). In addition, the scores of all dimensions in innovative behavior were significantly and positively correlated with organizational innovation climate ($r = 0.457$–$0.541$, $p < 0.01$), as shown in Table 3.

Table 2. The NIBS, NOICS, and STS scores for the Chinese clinical first-line nurses [n = 1,058, M (SD)].

| Dimensions | Number of items | Dimensional score | | Average score of items | | Ranking |
|---|---|---|---|---|---|---|
| | | M | SD | M | SD | |
| **NIBS total score** | 10 | 33.19 | 6.71 | 3.32 | 0.75 | — |
| Generating ideas | 3 | 10.14 | 2.83 | 3.38 | 0.79 | 1 |
| Obtaining support | 4 | 13.24 | 3.25 | 3.31 | 0.73 | 2 |
| Realizing ideas | 3 | 9.81 | 2.72 | 3.27 | 0.68 | 3 |
| **NOICS total score** | 21 | 68.88 | 12.76 | 3.28 | 0.83 | — |
| Organizational innovation incentives | 7 | 23.24 | 5.47 | 3.32 | 0.88 | 2 |
| Resource supply | 6 | 20.28 | 4.53 | 3.38 | 0.91 | 1 |
| Management practices | 8 | 25.36 | 6.26 | 3.17 | 0.83 | 3 |
| **STS total score** | 15 | 41.25 | 7.83 | 2.75 | 0.68 | — |

## 4.4. Correlation between innovative behavior and self-transcendence

Pearson's correlation analysis indicated that the total score of innovative behavior was significantly and positively correlated with self-transcendence ($r = 0.635$, $p < 0.01$). There was a significant and positive correlation between "Generating ideas," "Obtaining support," "Realizing ideas," and self-transcendence ($r = 0.573$–$0.622$, $p < 0.01$), as shown in Table 3.

## 4.5. Mediating effect of self-transcendence between innovative behavior and organizational innovation climate

The results of the mediating effect test indicated that the total impact of organizational innovation climate on innovative behavior was 0.362 ($p < 0.01$). 95% confidence interval (CI) was [0.321, 0.467]. The direct effect of organizational innovation climate on innovative behavior was 0.211 ($p < 0.01$), 95% CI was [0.265, 0.421]. The indirect effect value of self-transcendence on innovation behavior was 0.526×0.288 = 0.151, accounting for 41.7% of the total effect value of 0.362 ($p < 0.01$). The Boot lower limit CI to upper limit CI was [0.127, 0.313], which, excluding 0, indicates that the difference was statistically significant ($p < 0.05$), as shown in Table 4 and Fig 2.

Table 3. The correlations among innovative behavior, organizational innovation climate, and self-transcendence of the Chinese clinical first-line nurses (n = 1,058, r).

| Item | 1 | 1.1 | 1.2 | 1.3 | 2 | 2.1 | 2.2 | 2.3 | 3 |
|---|---|---|---|---|---|---|---|---|---|
| **1 NIBS total score** | — | | | | | | | | |
| 1.1 Generating ideas | 0.913** | — | | | | | | | |
| 1.2 Obtaining support | 0.908** | 0.902** | — | | | | | | |
| 1.3 Realizing ideas | 0.896** | 0.885** | 0.879** | — | | | | | |
| **2 NOICS total score** | 0.583** | 0.541** | 0.552** | 0.533** | — | | | | |
| 2.1 Organizational innovation incentives | 0.513** | 0.503** | 0.485** | 0.457** | 0.883** | — | | | |
| 2.2 Resource supply | 0.561** | 0.520** | 0.537** | 0.515** | 0.876** | 0.865** | — | | |
| 2.3 Management practices | 0.568** | 0.532** | 0.541** | 0.526** | 0.854** | 0.847** | 0.838** | — | |
| **3 STS total score** | 0.635** | 0.622** | 0.586** | 0.573** | 0.613** | 0.594** | 0.586** | 0.605** | — |

Note:

**: $p < 0.01$,

—: $r = 1$

**Table 4. The mediating effect of self-transcendence between innovative behavior and organizational innovation climate among the Chinese clinical first-line nurses (n = 1,058).**

| Model pathways | Standardized effect (*B*) | *SE* | *t*-value | *p*-value | *F* | *R* | *R²* | 95% *Cl* |
|---|---|---|---|---|---|---|---|---|
| **Total effect** | | | | | 425.624 | 0.625 | 0.391 | |
| Organizational innovation climate → Innovative behavior | 0.362 | 0.027 | 13.407 | < 0.001** | | | | [0.321, 0.467] |
| **Direct effect** | | | | | 255.308 | 0.588 | 0.346 | |
| Organizational innovation climate → Self-transcendence | 0.526 | 0.035 | 15.029 | < 0.001** | | | | [0.405, 0.502] |
| Organizational innovation climate → Innovative behavior | 0.211 | 0.018 | 11.722 | < 0.001** | | | | [0.265, 0.421] |
| Self-transcendence → Innovative behavior | 0.288 | 0.023 | 12.522 | < 0.001** | | | | [0.176, 0.308] |
| **Indirect effect** | | | | | — | — | — | |
| Organizational innovation climate → Self-transcendence → Innovative behavior | 0.151 | 0.016 | — | — | | | | [0.127, 0.313] |

Note:

**: $p < 0.01$

## 5. Discussion

### 5.1. Status quo of the innovative behavior, organizational innovation climate, and self-transcendence

In this study, the total score of innovative behavior of the clinical first-line nurses was 33.19 ± 6.71, which was moderate and consistent with the results conducted by Zhang et al. [3] and higher than those reported by Xu et al. [38]. The results indicated that the innovative behavior of the Chinese clinical first-line nurses during the Omicron pandemic needs further improvement. The reasons could be: (1) due to the pressure of the pandemic, the clinical first-line nurses experienced major psychological burdens, such as stress, anxiety, depression, sleep disturbances, and so on, which was considerably higher than the general population.

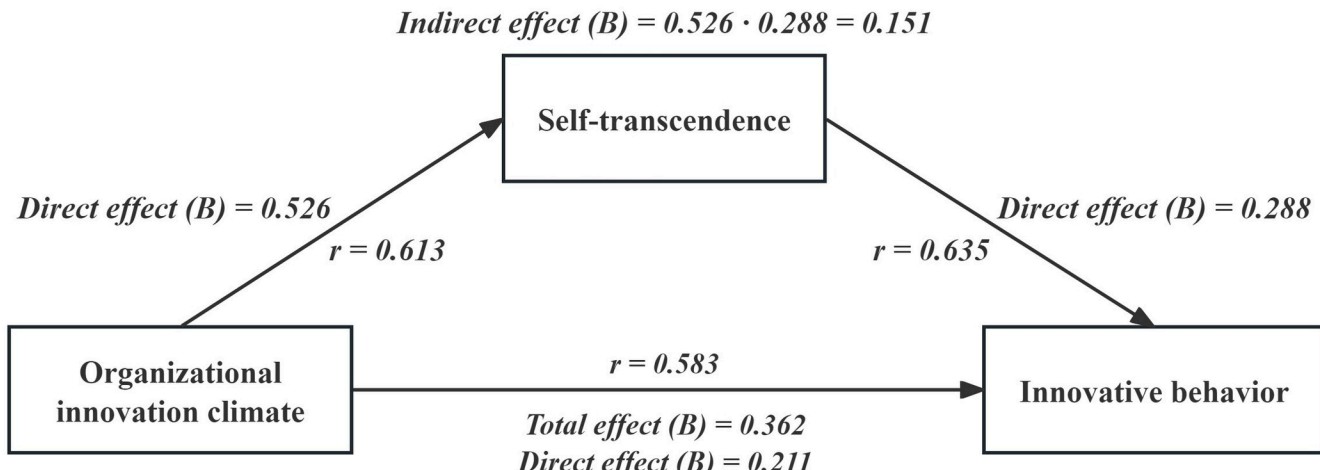

**Fig 2. The direct, indirect, and total effects between the innovative behavior, organizational innovation climate, and self-transcendence.**

Therefore, the nurses have shown a willingness to innovate and change their practices but not that high [9]; (2) the traditional nursing work modes and methods could not meet the needs of the pandemic situation, so the clinical first-line nurses were motivated to explore innovative ideas, improve their working mode, and adopt newer technologies and methods, which can improve the work efficiency and nursing quality, to meet the needs of the pandemic and better serve patients [39, 40]; (3) with the aging population in China, diversifying of diseases, and uneven distribution nursing resources, resulting in the traditional nursing service model that is unable to meet the personalized and dynamic needs of the patients for both quality and quantity [41]; (4) the National Health Planning Commission, the Ministry of Science and Technology, and six other departments in 2017 have successively issued the "Notice on Deepening the Reform Plan of the Medical and Health System in the Thirteenth Five-Year Plan," the "Special Plan for Health and Health Science and Technology Innovation in the Thirteenth Five-Year Plan," the "Central Committee of the Communist Party of China's Proposal on Formulating the Fourteenth Five-Year Plan for National Economic and Social Development" and the "Long-term Goals for the Third Five-Year Plan," which clarified the significance of scientific and technological innovation in the medical and health fields and emphasized the need to enhance medical care by strengthening the construction of the scientific and technological innovation system. Since then, the cultivation of innovative behavior of nurses has attracted the attention of hospitals, improving their innovative behavior [12].

Additionally, the dimension with the highest score was "Generating ideas," which was consistent with the results of Qiao et al. [42]. The study demonstrated that clinical first-line nurses can proactively identify and analyze issues, challenge conventional modes of thinking, and actively explore creative approaches and techniques to address the diverse challenges and requirements encountered during the clinical nursing process [43]. The dimension that received the lowest score was "Realizing ideas." This indicates that clinical first-line nurses may face limitations in their theoretical knowledge, scientific research capabilities, heavy workloads, and lack of departmental funding during the pandemic. As a result, they cannot receive the necessary support or assistance, leading to the premature termination of their innovative ideas and the subsequent failure of implementation [44, 45]. Nurse innovation behavior is an individual behavior in which nurses develop ideas that benefit patients, other nurses, departments, or hospitals and implement them in nursing work. It is also a type of behavior beyond the role that is typically not acknowledged by the organizational reward system [15]. Therefore, it is suggested that nursing managers should pay attention to the innovative ideas of nurses during clinical nursing work during the pandemic and should not stifle them at the stage of budding ideas. To increase their innovative enthusiasm and stimulate their innovative potential, they should foster a peaceful, innovative environment, encourage them to actively express their ideas, and listen to them carefully and patiently.

In this study, the total score of organizational innovation climate was 68.88 ± 12.76, similar to the result of Weng et al. [46]. The reason could be that during the Omicron pandemic, clinical first-line nurses frequently face not only physical harm but also psychological and spiritual suffering due to increased work demands and resource availability. The organizational innovation climate motivates people within the organization to generate innovative ideas. Management interventions to promote the organizational innovation climate can effectively stimulate work enthusiasm and initiative among nurses [20]. With a good nursing organization climate, the clinical first-line nurses can overcome negative emotions such as anxiety and fear caused by the Omicron pandemic as much as possible. This way, they are more likely to be recognized by colleagues and leaders, which is conducive to stimulating their enthusiasm for nursing and cultivating innovative behavior [21]. Nursing managers are advised to foster an organizational climate that promotes innovation among clinical first-line nurses. This can be achieved by

enhancing scientific research training and facilitating their participation in research and innovation training activities. Additionally, creating a harmonious and friendly team environment will further encourage innovation.

The study found that the self-transcendence score was 41.25 ± 7.83, indicating a moderate level. This score was comparable to the findings of Zhang et al. [47] but significantly higher than the score reported by Zhao et al. [48] for undergraduate nursing students. However, it was lower than the score reported by Zhu et al. [49] for nurses. Self-transcendence manifests cognition and conation, which involves many aspects, such as psychology, spirit, and mind. Moreover, it is related to the self-spiritual acceptance and individual recognition experience [28]. Self-transcendence is also a powerful coping mechanism that can help individuals adapt to physical, emotional, and spiritual pain, thus promoting happiness [50]. Palmer et al. [29] also found that nurses with higher self-transcendence have more work-related vitality, dedication, and concentration, a stronger sense of job involvement, and better innovative behavior and ability. The low participation rate of clinical first-line nurses in scientific research training, publication of academic papers, patent applications, and research topic applications in this study may hinder the development of their innovative spirit and abilities. Consequently, it may impede the realization of their personal values and fulfillment of their self-realization needs, thereby limiting the enhancement of nurses' self-transcendence level. Compared to intern nurses, clinical first-line nurses can contact patients with many abilities in clinical nursing. When facing the patients' disease pains and nursing needs, the clinical first-line nurses will actively expand their nursing knowledge and skills in many ways to achieve a higher level of self-realization and constantly surpass the status quo to meet the overall needs of patients. It is suggested that nursing managers should promote the mental health of clinical first-line nurses by improving their clinical practice and team communication abilities during the pandemic, promoting nursing management reformation, and spreading evidence-based practice to enhance their self-transcendence level [51].

## 5.2. Positive correlation between innovative behavior and organizational innovation climate

The results indicated that innovative behavior was significantly and positively correlated with organizational innovation climate ($r = 0.583$, $p < 0.01$), meaning the better the organizational innovation climate, the better the innovative behavior of the clinical first-line nurses, which was consistent with the studies by Zhang et al. [3] and Lv et al. [41]. The organizational innovation climate of nurses pertains to their ongoing cognitive perception of the level of support for innovation within hospital environments. This external factor significantly influences the implementation of innovative nursing practices [52]. If the organizational innovation climate of nurses is improved, hospitals will be more content with the resources provided to nurses, the incentives for organizational innovation, and the management practices. Additionally, there is greater support regarding materials, funds, relevant policies, attention, and management of innovative practice activities. This provides a crucial foundation and assurance for nurses to engage in innovative activities and practical scientific research, increasing their innovative behavior [46]. It is suggested that nursing managers should pay more attention and actively create an excellent internal and external organizational innovation climate during the pandemic. By implementing novel practice initiatives, organizations can establish a conducive environment that fosters innovation. Managers can provide financial support for nurses' scientific research and innovative training, develop a system of rewards and punishment, encourage internal innovation consciousness and enthusiasm, stimulate the innovative potential of nurses, and enhance their innovative behavior.

### 5.3. Positive correlation between innovative behavior and self-transcendence

There was a significant and positive correlation between the nurses' innovative behavior and self-transcendence ($r = 0.635$, $p < 0.01$), implying that the higher the self-transcendence, the better the innovative behavior of clinical first-line nurses, similar to the study by Zhang et al. [47]. Self-transcendence is a specific experience, expanding the boundaries of personal ability in many ways, constantly surpassing the status quo to achieve a higher level of self-realization, and attain harmony and unity with nature, others, and self [50]. Studies showed that a nurse with good self-transcendence has better innovative behavior with greater work-related vitality, dedication, concentration, a stronger sense of job involvement, a better ability to reflect their value, and meet their self-realization needs [53]. It is suggested that nursing managers should improve the self-transcendence level of clinical first-line nurses in multiple aspects and directions by taking incentive measures to promote their innovative behavior.

### 5.4. A mediating role of self-transcendence between innovative behavior and organizational innovation climate

The results demonstrated that self-transcendence partially mediated between innovative behavior and organizational innovation climate, accounting for 41.7% ($p < 0.01$). Nursing managers should prioritize the self-transcendence level of clinical first-line nurses and mediate between their innovative behavior and the organizational innovation climate [54]. This approach can enhance the organizational innovation climate for clinical first-line nurses, directly improving their innovative behavior. Additionally, it can indirectly enhance their innovative behavior by enhancing their self-transcendence level. Nursing managers should also be fully empowered to foster an environment of love, tolerance, innovation, and competition. Clinical first-line nurses should be encouraged to conduct self-examination and reflection against clinical nursing, given full rein to their subjective initiative and opportunities and platforms for their professional growth and development. In addition, the corresponding reward and punishment mechanism can be set up during the Omicron pandemic, aiming to mobilize their inner innovation consciousness and enthusiasm, stimulate their innovation potential, and improve their innovation behavior.

### 5.5. Limitations

There were some limitations in this study. First, the study was conducted using a convenience sampling method. Only 1,058 Chinese clinical first-line nurses were enrolled from seven tertiary grade-A hospitals, which were all located in Tianjin city in Northern China. This may lead to unrepresentative samples and one-sided, non-generalized, or limited results. Therefore, random, multi-center research with a large sample should be conducted in the future.

Second, the study only used an online questionnaire, which may have disadvantages in the accuracy of the questionnaire responses and on-site quality control. Thus, online and offline surveys should be adopted to reduce bias in the future.

Third, the study did not employ multiple linear regression analysis to examine the factors influencing innovative behavior, organizational innovation climate, and self-transcendence among nurses. Furthermore, the data analysis did not take gender differences in the sample into account. Additional investigation is needed to enhance the rigor of the design. It is recommended that future studies incorporate a greater number of clinical first-line nurses from various regions, delve into the factors that influence innovation behavior among these nurses, and account for gender disparities within the sample.

Fourth, the statistical model may have overlooked potential confounders other than "organizational innovation climate" and "self-transcendence" that might significantly affect "innovative behavior." Therefore, more accurate and scientific statistical models must be adopted in future studies, along with more factors that may significantly affect innovation behavior.

### 5.6. Implications for nursing management

This study can provide a theoretical basis for constructing the intervention measures to improve the innovative behavior and ability of the Chinese clinical first-line nurses, further improving the nursing quality and satisfaction during the Omicron pandemic, thereby promoting the development of clinical nursing. This study demonstrated that the organizational innovation climate could affect the innovative behavior among the Chinese clinical first-line nurses during the Omicron pandemic directly and indirectly through the mediating role of self-transcendence. Nursing managers should use suitable management styles and methods for each nurse during the pandemic, create an open and positive work environment for innovation, encourage nurses to be enthusiastic about and aware of new ideas, increase the quality and quantity of information they get, help nurses deal with the negative emotions caused by the pandemic, and then encourage their innovative behavior to make nursing better.

## 6. Conclusion

This study found that the innovative behavior, organizational innovation climate, and self-transcendence among the 1,058 Chinese clinical first-line nurses during the Omicron pandemic were moderate and needed improvement. In addition, there were significant and positive correlations between innovative behavior, organizational innovation climate, and self-transcendence. Self-transcendence played a mediating role in the change in innovative behavior and the organizational innovation climate. During the pandemic, nursing managers can make the workplace a more welcoming and cooperative environment for innovation, establish a scientific and varied training program to boost nurses' creative thinking, increase the quantity and quality of information available on innovation, assist nurses in overcoming negative emotions such as fear, raise their level of self-transcendence, and grow their creative enthusiasm and awareness while performing clinical work.

## Supporting information

**S1 File. STROBE checklist.**
(DOCX)

**S2 File. Dataset used for the study.**
(XLSX)

**S1 Appendix. Questionnaires for the study.**
(DOCX)

## Acknowledgments

All researchers would like to express our gratitude to all the participants for taking their precious time to participate in this study, and also thank the hospital managers and nursing administrators for their strong support and help to this study. Furthermore, we would like to thank KetengEdit (www.ketengedit.com) for its linguistic assistance during the preparation of this manuscript.

## Author Contributions

**Conceptualization:** Zhangyi Wang, Yue Zhu, Huifang Zhou, Xiaoli Pang, Xiaochun Tang, Li Liu.

**Data curation:** Zhangyi Wang, Yue Zhu, Tingrui Wang, Huifang Zhou, Lamei Chen.

**Formal analysis:** Zhangyi Wang, Tingrui Wang, Tao Su, Siai Zhang, Mengru Liu, Lamei Chen, Manli Wu, Liping Li, Xuechun Li, Jiaofeng Peng, Li Liu.

**Funding acquisition:** Zhangyi Wang, Yue Zhu, Xiaochun Tang.

**Investigation:** Yue Zhu, Tingrui Wang, Tao Su, Huifang Zhou, Siai Zhang, Mengru Liu, Lamei Chen, Manli Wu, Liping Li, Xuechun Li, Li Liu.

**Methodology:** Tingrui Wang, Tao Su, Manli Wu, Xiaoli Pang, Jiaofeng Peng, Xiaochun Tang, Li Liu.

**Project administration:** Zhangyi Wang, Yue Zhu, Xiaoli Pang.

**Resources:** Zhangyi Wang, Tingrui Wang, Huifang Zhou, Mengru Liu, Li Liu.

**Software:** Mengru Liu.

**Supervision:** Huifang Zhou.

**Validation:** Zhangyi Wang, Liping Li, Jiaofeng Peng.

**Visualization:** Zhangyi Wang, Tingrui Wang, Tao Su, Siai Zhang, Li Liu.

**Writing – original draft:** Zhangyi Wang, Yue Zhu, Huifang Zhou, Siai Zhang, Lamei Chen, Jiaofeng Peng, Xiaochun Tang.

**Writing – review & editing:** Zhangyi Wang, Yue Zhu, Xuechun Li, Jiaofeng Peng, Li Liu.

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
