## [Decision Letter · Decision Letter 0]

19 Feb 2024

PONE-D-23-28215Innovative behaviour and organizational innovation climate among Chinese clinical first-line nurses during the Omicron pandemic: The mediating roles of self-transcendencePLOS ONE

Dear Dr. Tang,

Thank you for submitting your manuscript to PLOS ONE. After careful consideration, we feel that it has merit but does not fully meet PLOS ONE’s publication criteria as it currently stands. Therefore, we invite you to submit a revised version of the manuscript that addresses the points raised during the review process.

We look forward to receiving your revised manuscript.

Kind regards,

Myriam M. Altamirano-Bustamante

Academic Editor

PLOS ONE

Journal Requirements:

   "This study was supported by the Tianjin Research Innovation Project for Postgraduate Students (CN) [Grant numbers: 2021YJSS171], the Tianjin University of Traditional Chinese Medicine Science and Technology Innovation Fund Project for College Students (CN) [Grant numbers: ZX01], and the Tianjin University of Traditional Chinese Medicine Research Innovation Project for Postgraduate Students (CN) [Grant numbers: YJSKC-20212005]." 

5. In the online submission form, you indicated that "The relevant data of this study can be obtained from the first author (283537548@qq.com) or corresponding author (2995620556@qq.com) on reasonable request."

Reviewers' comments:

Reviewer's Responses to Questions

**Comments to the Author**

1. Is the manuscript technically sound, and do the data support the conclusions?

Reviewer #1: Yes

2. Has the statistical analysis been performed appropriately and rigorously? 

Reviewer #1: N/A

3. Have the authors made all data underlying the findings in their manuscript fully available?

Reviewer #1: Yes

4. Is the manuscript presented in an intelligible fashion and written in standard English?

Reviewer #1: Yes

5. Review Comments to the Author

Reviewer #1: The manuscript describes an interesting study done on the first-line clinical nurses who worked during the Omicron pandemic in general and specialized hospitals in China. The background of the study is very thorough in describing the challenges faced by nurses during the pandemic including the risk of contracting Omicron mutant strain of the COVID-19, but also the mental and emotional problems faced by the increased workload such as depression, anxiety and post-traumatic stress disorders.

The pandemic forced nurses to innovate in the work organization, techniques and the roles played inside the hospitals. The good results, quality of attention and improved prognosis offered to patients when nurses went on to innovate in these areas, encouraged the authors to specifically outline the role of innovative behavior in the clinical setting, looking into the factors that favor innovation. Hence, organizational innovation and self-transcendence are taken into consideration to see their role and influence in innovative behavior.

The study has a convenience sample, properly declared in the limitations section, of 1058 nurses who answered the 4 questionnaires: the sociodemographic questionnaire, the Nurse Innovative behavior scale, the Nurse Organizational Innovation Climate Scale and Self-Transcendence Scale. All of which had proper ethical provision taken to obtain data from participants.

The results section presents very neatly the scores in each of the questionnaires and scales and shows the correlations between innovative behavior and organizational innovations climate, as well as between innovative behavior and self-transcendence; a mediating effect test is performed to indicate an indirect effect of self-transcendence between innovative behavior and organizational innovation climate.

The discussion turn out to be very interesting and is well supported on the data reviewed in the background of the study, as well as, the empirical data obtained from the participants. A well treated discussion is undertaken concerning the need of innovative behavior to face the great clinical challenges that rose during the Omicron pandemic. Regular nursing methods and techniques would have not been sufficient to meet this challenge, so there is a suggestion for nursing managers to enhance the innovation capabilities of the nursing staff because their scores in innovative behavior were moderate. A favorable organizational innovation climate stimulates positive responses and attitudes among nurses as found in the study´s results, but also concurring with other relevant literature. A positive correlation was found between these two results.

However, self-transcendence seem to be a more complex variable to take into consideration, it is considered to have relation with the vitality, dedication and the motivations to keep a high-spirited attention to their work and patients. A special and interesting discussion is developed by the authors concerning the role of academic and scientific activities in keep self-transcendence high in nurses. The discussion of the mediating role of self-transcendence between innovative behavior and organizational innovative climate is interesting because it hints possible paths towards improving the latter two.

Considering these remarks on the subject, I would like to make some annotations to specific issues in the manuscript that I believe should be taken into consideration:

1. In the definition of innovative behavior the authors start form the definition of Scott et al [19] which is a wide ranging definition, and later on cite some of the specific literature, but some examples of what specifically are they are conceptualizing as innovative behavior in the Chinese clinical setup would be welcome.

2. When laying out the Space Theory to express a concrete formula for innovative behavior (Bi= f(Pi•Ei)), it seems that the factor taken into consideration in this formula are wider than the ones taken in this study, p.e. Pi which considers personal internal needs of psychological factors can be considered much wider than the results expressed in the Nurse Innovative behavior scale and the Self-Transcendence Scale. The same goes for Ei and the Nurse Organizational Innovation Climate Scale. Therefore, a brief explanation of how the trim is justified would underscore this step, considering that measurement of such wide elements would be very hard.

3. Being that the sample is mostly women nurses, when discussing self-transcendence there is no consideration on gender and what could this complex concept mean to women or if some of the data could be disaggregated in this respect.

6. PLOS authors have the option to publish the peer review history of their article (what does this mean?). If published, this will include your full peer review and any attached files.

Reviewer #1: No

---

## [Author Response · Author response to Decision Letter 0]

21 Feb 2024

February 22th, 2024

Dear academic editor and reviewers:

 We would like to thank you for your efforts in reviewing our manuscript entitled “Innovative behaviour and organizational innovation climate among Chinese clinical first-line nurses during the Omicron pandemic: The mediating roles of self-transcendence”, and providing many helpful comments and suggestions, which will all prove invaluable in the revision and improvement of our paper, as well as in guiding our research in the future.

 According to your nice suggestions, we have studied your comments point-by-point, and have made extensive corrections to our previous draft, the detailed corrections are listed below. All authors have approved the response letter and the revised version of the manuscript. 

Responds to the reviewers’ comments point-by-point:

Response to reviewer #1: 

 Dear reviewer, we appreciate it very much for these good suggestion. Additionally, we also thank you for praising our manuscript is generally well-written. We are well aware that there are still some shortcomings in the manuscript. Therefore, in accordance with your suggestions, we have made the following revisions to our manuscript: . 

 Q1: “In the definition of innovative behavior the authors start form the definition of Scott et al [19] which is a wide ranging definition, and later on cite some of the specific literature, but some examples of what specifically are they are conceptualizing as innovative behavior in the Chinese clinical setup would be welcome.

 Response: Thank you very much for your valuable comments, for this suggestion, we have made some revisions. We have further elaborated the definition of nurses' innovative behavior in China, which has the characteristics and cultural background of China, which is “Besides, Bao's [24] definition of nurses' innovative behavior is the most widely accepted in China: In order to promote health, prevent diseases and improve the quality of patients' care, nurses seek and develop new methods, technologies and working methods, and after obtaining the support of others, introduce and apply them to the behavioral process at work, including three stages: generating ideas, obtaining support and realizing ideas.” And all the changes have been marked in red in the revised draft, But it may not be so good. And our writing ability is limited, I hope you can be magnanimous.

 Q2: “When laying out the Space Theory to express a concrete formula for innovative behavior (Bi= f (Pi•Ei)), it seems that the factor taken into consideration in this formula are wider than the ones taken in this study, p.e. Pi which considers personal internal needs of psychological factors can be considered much wider than the results expressed in the Nurse Innovative behavior scale and the Self-Transcendence Scale. The same goes for Ei and the Nurse Organizational Innovation Climate Scale. Therefore, a brief explanation of how the trim is justified would underscore this step, considering that measurement of such wide elements would be very hard.”

 Response: Thank you very much for your valuable comments, for this suggestion, we have made some revisions. When we introduce the definitions of Innovative behaviour, organizational innovation climate and self-transcendence in the preface, we further elaborate their respective attribute characteristics. Of course, the range of variables included in Bi, Pi and Ei of the Space Theory is indeed very wide, and we also refer to the variables included in previous similar studies, which may be biased to some extent, and is also a deficiency of this study. And all the changes have been marked in red in the revised draft, But it may not be so good. And our writing ability is limited, I hope you can be magnanimous.

 Q3: “Being that the sample is mostly women nurses, when discussing self-transcendence there is no consideration on gender and what could this complex concept mean to women or if some of the data could be disaggregated in this respect.”

 Response: Thank you very much for your valuable comments, for this suggestion, we want to explain that: Because this study is a correlation study, plus the requirements of the number of words and the layout of the article. Therefore, in the data analysis, the three variables are not analyzed by single factor and multiple linear regression respectively, and only the status quo and reasons of the three variables are analyzed in the discussion part, and the influencing factors of the three variables are not analyzed separately. So, there is no consideration on gender and what could this complex concept mean to women when discussing self-transcendence. This is also a deficiency of this study, which is also described in the section "5.5 | Limitations". And all the changes have been marked in red in the revised draft, But it may not be so good. And our writing ability is limited, I hope you can be magnanimous.

 Last but not least, we have studied your comments point by point, we tried our best to improve the manuscript and made some changes to the manuscript. These changes will not influence the content and framework of the paper. And here we did not list the changes in detail but marked in red in the revised paper. We appreciate for your warm work earnestly and hope that the correction will meet with approval. All authors have approved the response letter and the revised version of the manuscript. 

 And due to our writing ability, understanding and logic are limited and we look forward to your generosity. Due to the time limit of our project and the need to employ professional title, and this paper is very important to our team, so we sincerely hope that you can recommend our revised version of the manuscript to accept for publication in the journal Plos One as soon as possible. If you have any queries, please do not hesitate to contact me. 

 Thank you again for your valuable comments and suggestions. I'm looking forward to hearing from you soon in due course.

 Best regards,

 Zhangyi Wang, Master's degree, Nurse Practitioner, 283537548@qq.com, 

 Nursing Department, Affiliated Hengyang Hospital of Hunan Normal University & Hengyang Central Hospital, Hengyang City, Hunan Province, 421001, China.

---

## [Decision Letter · Decision Letter 1]

3 Apr 2024

PONE-D-23-28215R1Innovative behaviour and organizational innovation climate among Chinese clinical first-line nurses during the Omicron pandemic: The mediating roles of self-transcendencePLOS ONE

Dear Dr. Wang,

Thank you for submitting your manuscript to PLOS ONE. After careful consideration, we feel that it has merit but does not fully meet PLOS ONE’s publication criteria as it currently stands. Therefore, we invite you to submit a revised version of the manuscript that addresses the points raised during the review process.

We look forward to receiving your revised manuscript.

Kind regards,

Myriam M. Altamirano-Bustamante

Academic Editor

PLOS ONE

Reviewers' comments:

Reviewer's Responses to Questions

**Comments to the Author**

1. If the authors have adequately addressed your comments raised in a previous round of review and you feel that this manuscript is now acceptable for publication, you may indicate that here to bypass the “Comments to the Author” section, enter your conflict of interest statement in the “Confidential to Editor” section, and submit your "Accept" recommendation.

Reviewer #1: (No Response)

Reviewer #2: (No Response)

2. Is the manuscript technically sound, and do the data support the conclusions?

Reviewer #1: Partly

Reviewer #2: Partly

3. Has the statistical analysis been performed appropriately and rigorously? 

Reviewer #1: N/A

Reviewer #2: No

4. Have the authors made all data underlying the findings in their manuscript fully available?

Reviewer #1: Yes

Reviewer #2: No

5. Is the manuscript presented in an intelligible fashion and written in standard English?

Reviewer #1: Yes

Reviewer #2: Yes

6. Review Comments to the Author

**Reviewer #1:** I thank the authors for the changes made to the manuscript in attention to the previous observations. The manuscript has improved where it made those changes. I appreciate the attention paid to question 1., and the further details provided by Bao’s definition of nurses innovative behavior. In this regard, I consider the matter resolved.

Concerning Q3, I think that not taking in consideration the matter of gender in such a sample is a limitation, but of course, the authors may legitimately choose to keep the study in such fashion and not consider analyzing the data considering differences between genders, however this could be explicitly stated in the limitations section.

It is in Q2 where I found some difficulty to review the response of the authors, as they say “the range of variables included in Bi, Pi and Ei of the Space Theory is indeed very wide, and we also refer to the variables included in previous similar studies, which may be biased to some extent, and is also a deficiency of this study.” But when trying to trace how the other studies defined the Bi, Pi or Ei variables, I wasn’t able to read those study as they were mostly written in Chinese (language that I am unable to read). Therefore, I’m unable to follow the argument presented and therefore cannot judge the sufficiency of the changes made or the antecedents put forward, perhaps the argument should be made explicit in the text.

Consider correcting the following: pg. 25 “Innovation behaviour.among clinical first-line nurses.”

**Reviewer #2:** REVIEW

This study delves into the interrelationships among "innovative behavior," "organizational innovation climate," and "self-transcendence" in front-line clinical nurses in China during the Omicron pandemic, underpinned by field theory. Using a cross-sectional design, it incorporates a convenience sample of 1,058 nurses from seven hospitals and assesses the mentioned constructs through validated scales. The statistical analysis focuses on these dimensions to explore their correlations and the mediating role of "self-transcendence" between "organizational innovation climate" and "innovative behavior."

While it's crucial to gather data on nurses' experiences in Chinese hospitals during the Covid-19 pandemic to assess and improve nursing in China under crisis conditions, this draft has several areas that require rectification for potential publication in PLOS ONE.

MAJOR ISSUES

1. Samples: The mere mention of "seven hospitals in China" without details on their location, size, or administrative level significantly undermines the study's generalizability. Moreover, while nurse characteristics are described, information on the hospitals is notably absent but essential.

2. Sample Size: The method described for calculating sample size is incorrectly attributed to Kendall's "Multivariate Analysis," confusing the number of items with variables. The approach is inappropriate for this study's aims, specifically concerning the precision of correlation coefficient estimation. Moreover, the cited page range (251-306) does not exist in Kendall's work, which casts doubt on the draft's accuracy.

3. Theoretical Foundation: Although leveraging field theory is intriguing, its application to the authors' mediating variable model is inadequately justified and discussed.

4. Statistical Model: The core model may be overlooked potential confounders other than "organizational innovation climate" and "self-transcendence" that might significantly affect "innovative behavior."

5. Table 1: The inclusion of P-values and the unclear meaning of "Z/H" are problematic. Descriptive information should not intertwine with inferential statistics, contrary to what STROBE guidelines advocate (Item 14, Explanation).

6. Mediating Effect of Self-Transcendence: This section and Figure 2 inadequately report the direct effects between "organizational innovation climate" and "innovative behavior," requiring a clearer presentation in both figure and table formats.

MINOR ISSUES

1. References: Several citations are incomplete or incorrect, such as the nonexistent "Review of Principles of Topological Psychology" and the misquoted page range in Kendall's work. Additionally, the "PROCESS" program used for mediation analysis is not cited. A thorough review and correction of the reference list are recommended.

7. PLOS authors have the option to publish the peer review history of their article (what does this mean?). If published, this will include your full peer review and any attached files.

Reviewer #1: No

Reviewer #2: **Yes: **Chiharu Murata

---

## [Author Response · Author response to Decision Letter 1]

9 Apr 2024

Submission ID: PONE-D-23-28215 R2

Title: Innovative behavior and organizational innovation climate among the Chinese clinical first-line nurses during the Omicron pandemic: The mediating roles of self-transcendence 

April 9th, 2024

Dear academic editor and reviewers:

 We would like to thank you for your efforts in reviewing our revised manuscript entitled “Innovative behavior and organizational innovation climate among the Chinese clinical first-line nurses during the Omicron pandemic: The mediating roles of self-transcendence”, and providing many helpful comments and suggestions, which will all prove invaluable in the revision and improvement of our paper, as well as in guiding our research in the future.

 According to your nice suggestions, we have studied your comments point-by-point, and have made extensive corrections to our previous draft, and all the changes have been marked in red in the revised draft, the detailed corrections are listed below. In order for reviewers and readers to better understand and read, we invited two nursing professors whose mother tongue is English and KetengEdit (www.ketengedit.com) to polish and revise the language of our article. All authors have approved the response letter and the revised version of the manuscript. 

Responds to the reviewers’ comments point-by-point:

Response to reviewer #1: 

 Dear reviewer, we appreciate it very much for these good suggestion. Additionally, we also thank you for praising our manuscript is generally well-written. We are well aware that there are still some shortcomings in the manuscript. Therefore, in accordance with your suggestions, we have made the following revisions to our manuscript: 

Q1: “I thank the authors for the changes made to the manuscript in attention to the previous observations. The manuscript has improved where it made those changes. I appreciate the attention paid to question 1., and the further details provided by Bao’s definition of nurses innovative behavior. In this regard, I consider the matter resolved.”

 Response: For this, thank you very much for your praise.

Q2: “Concerning Q3, I think that not taking in consideration the matter of gender in such a sample is a limitation, but of course, the authors may legitimately choose to keep the study in such fashion and not consider analyzing the data considering differences between genders, however this could be explicitly stated in the limitations section.”

 Response: Thank you very much for your valuable comments, for this suggestion, we have made some revisions. We also think that not taking in consideration the matter of gender in such a sample is a limitation in this study, which is also described in the section "5.5. Limitations", that is, “ Third, the study did not employ multiple linear regression analysis to examine the factors influencing innovative behavior, organizational innovation climate, and self-transcendence among nurses. Furthermore, the data analysis did not take gender differences in the sample into account. Additional investigation is needed to enhance the rigor of the design. It is recommended that future studies incorporate a greater number of clinical first-line nurses from various regions, delve into the factors that influence innovation behavior among these nurses, and account for gender disparities within the sample.” And all the changes have been marked in red in the revised draft, But it may not be so good. And our writing ability is limited, I hope you can be magnanimous.

Q3: “It is in Q2 where I found some difficulty to review the response of the authors, as they say “the range of variables included in Bi, Pi and Ei of the Space Theory is indeed very wide, and we also refer to the variables included in previous similar studies, which may be biased to some extent, and is also a deficiency of this study.” But when trying to trace how the other studies defined the Bi, Pi or Ei variables, I wasn’t able to read those study as they were mostly written in Chinese (language that I am unable to read). Therefore, I’m unable to follow the argument presented and therefore cannot judge the sufficiency of the changes made or the antecedents put forward, perhaps the argument should be made explicit in the text.”

 Response: Thank you very much for your valuable comments, for this suggestion, we have made some revisions. First of all, we are sorry that this theory was put forward by Zhang Liming, a scholar in China, so we may not be able to read relevant English articles. Therefore, we further explain the argument of this theory, namely, “The theoretical framework of this study was Space Theory, proposed according to the field dynamics theory. According to the theory of field dynamics, the formula of innovation behavior is Bi = f·(Si), where innovation behavior (Bi) is a function of the innovation space, Si. According to this theory, innovation, as a social behavior, also has an innovative situation, and this situation is the innovative space. Therefore, another form of innovative behavior formula is finally obtained: Bi = f (Pi·Ei), where Bi stands for behavior (including innovative behavior), Pi stands for the person (including personal internal needs, internal psychological factors, and so forth), and Ei stands for the environment (including innovative environment and external environmental factors). Furthermore, it is important to note that the innovation subject Pi and the innovation environment Ei are interdependent variables that comprise Si. Ei can be understood as the sum of social and natural relations affecting Bi and Si, which can be regarded as the dynamic field of innovation behavior. In Si, Pi and Ei will interact. Based on the theories of space and field dynamics, the innovative behavior of nurses can be explained as a response to external factors that disrupt their usual working state. Nurses engage in innovative behavior to adapt to their environment, restore balance, and continue their work in nursing. During the Omicron pandemic, clinical first-line nurses must ensure the quality of care while facing greater work pressure, which may give birth to innovative behavior of nurses. Furthermore, this demonstrates how external factors impact the internal factors of nurses and promote their innovative behavior. In particular, combined internal and external factors can affect the innovative behavior of nurses, and may affect the innovative behavior of nurses through the intermediary role of internal factors.” 

 And further modify that conceptual framework diagram of Figure 1 in this study. What’s more, when we introduce the definitions of innovative behavior, organizational innovation climate and self-transcendence in the preface, we further elaborate their respective attribute characteristics. And all the changes have been marked in red in the revised draft, but it may not be so good. And our writing ability is limited, I hope you can be magnanimous.

Q4: “Consider correcting the following: pg. 25 “Innovation behaviour.among clinical first-line nurses.”

 Response: Thank you very much for your valuable comments, for this suggestion, we have changed it to “Innovation behaviour among clinical first-line nurses.”

Response to reviewer #2: 

 Dear reviewer, we appreciate it very much for these good suggestion. We are well aware that there are still some shortcomings in the manuscript. Therefore, in accordance with your suggestions, we have made the following revisions to our manuscript: 

Q1: “1. Samples: The mere mention of "seven hospitals in China" without details on their location, size, or administrative level significantly undermines the study's generalizability. Moreover, while nurse characteristics are described, information on the hospitals is notably absent but essential.”

 Response: Thank you very much for your valuable comments, for this suggestion, we have made some revisions. We have described and supplemented the hospital information in the section of “3.2. Participants and sample” and the section of “4.1. Demographic characteristics”, that is, “Convenience sampling was used to recruit the nurses from seven tertiary grade-A hospitals during the Omicron pandemic, which are all located in a municipality directly under the central government in Northern China, with more than 1,500 beds and 2,000 employees.” “A total of 1,073 Chinese clinical first-line nurses were recruited to participate in the survey from seven tertiary grade-A hospitals located in a municipality directly under the central government in Northern China, with more than 1,500 beds and 2,000 employees. Among them, there are five general and two specialty hospitals.”, respectively. But it may not be so complete, which is also a deficiency of this study. Moreover, due to the blind review of the Plos One and the limitation of hospital privacy, we failed to describe the specific names and cities of seven hospitals, but only described their nature, size, number of hospital beds and number of employees, etc.We also think the sample of this study is a limitation in this study, therefore, it is also described in the section "5.5. Limitations", that is, “First, the study was conducted using a convenience sampling method. Only 1,058 Chinese clinical first-line nurses were enrolled from seven tertiary grade-A hospitals, which were all located in a municipality under the central government in Northern China. This may lead to unrepresentative samples and one-sided, non-generalized, or limited results. Therefore, random, multi-center research with a large sample should be conducted in the future.” And all the changes have been marked in red in the revised draft, but it may not be so good. And our writing ability is limited, I hope you can be magnanimous.

Q2: “2. Sample Size: The method described for calculating sample size is incorrectly attributed to Kendall's "Multivariate Analysis," confusing the number of items with variables. The approach is inappropriate for this study's aims, specifically concerning the precision of correlation coefficient estimation. Moreover, the cited page range (251-306) does not exist in Kendall's work, which casts doubt on the draft's accuracy.”

 Response: Thank you very much for your valuable comments, for this suggestion, we have made some revisions. We have changed “Kendall” to “G*Power 3.1.9.7 software”, that is, “G*Power 3.1.9.7 software was used to calculate the minimum sample size required for correlation analysis (α error probability was 0.05, and test validity was 80%). According to related research, the average correlation coefficient among innovative behavior, organizational innovation climate, and self-transcendence is 0.17, and the sample size is calculated using a t-test. In G*Power 3.1.9.7 software, the t-tests method was selected, and "Correlation: Point biserial model" was selected, and then the parameter effect size |ρ| = 0.17, α = 0.05, β = 0.8 were set, and finally clicked "Calculate" to calculate the total sample size = 266, the minimum sample size of this study. In the end, 1, 058 participants were included in this study.” And all the changes have been marked in red in the revised draft, But it may not be so good. And our writing ability is limited, I hope you can be magnanimous.

Q3: “3. Theoretical Foundation: Although leveraging field theory is intriguing, its application to the authors' mediating variable model is inadequately justified and discussed.”

 Response: Thank you very much for your valuable comments, for this suggestion, we have made some revisions. We further explain the argument of this theory, namely, “The theoretical framework of this study was Space Theory, proposed according to the field dynamics theory. According to the theory of field dynamics, the formula of innovation behavior is Bi = f·(Si), where innovation behavior (Bi) is a function of the innovation space, Si. According to this theory, innovation, as a social behavior, also has an innovative situation, and this situation is the innovative space. Therefore, another form of innovative behavior formula is finally obtained: Bi = f (Pi·Ei), where Bi stands for behavior (including innovative behavior), Pi stands for the person (including personal internal needs, internal psychological factors, and so forth), and Ei stands for the environment (including innovative environment and external environmental factors). Furthermore, it is important to note that the innovation subject Pi and the innovation environment Ei are interdependent variables that comprise Si. Ei can be understood as the sum of social and natural relations affecting Bi and Si, which can be regarded as the dynamic field of innovation behavior. In Si, Pi and Ei will interact. Based on the theories of space and field dynamics, the innovative behavior of nurses can be explained as a response to external factors that disrupt their usual working state. Nurses engage in innovative behavior to adapt to their environment, restore balance, and continue their work in nursing. During the Omicron pandemic, clinical first-line nurses must ensure the quality of care while facing greater work pressure, which may give birth to innovative behavior of nurses. Furthermore, this demonstrates how external factors impact the internal factors of nurses and promote their innovative behavior. In particular, combined internal and external factors can affect the innovative behavior of nurses, and may affect the innovative behavior of nurses through the intermediary role of internal factors.” 

 And the conceptual framework and theoretical basis of Figure 1 in this study are further modified and explained. What’s more, when we introduce the definitions of innovative behavior, organizational innovation climate and self-transcendence in the preface, we further elaborate their respective attribute characteristics. And all the changes have been marked in red in the revised draft, but it may not be so good. And our writing ability is limited, I hope you can be magnanimous.

Q4: “Statistical Model: The core model may be overlooked potential confounders other than "organizational innovation climate" and "self-transcendence" that might significantly affect "innovative behavior".”

 Response: Thank you very much for your valuable comments, for this suggestion, we also agree with the limitations of the core statistical model. Therefore, it is also described in the section "5.5. Limitations", that is, “Fourth, the statistical model may have overlooked potential confounders other than "organizational innovation climate" and "self-transcendence" that might significantly affect "innovative behavior." Therefore, more accurate and scientific statistical models must be adopted in future studies, along with more factors that may significantly affect innovation behavior.” And all the changes have been marked in red in the revised draft, but it may not be so good. And our writing ability is limited, I hope you can be magnanimous.

Q5: “Table 1: The inclusion of P-values and the unclear meaning of "Z/H" are problematic. Descriptive information should not intertwine with inferential statistics, contrary to what STROBE guidelines advocate (Item 14, Explanation).”

 Response: Thank you very much for your valuable comments, for this suggestion, we have made some revisions in Table 1. Because the aims of this study are (1) to investigate the innovative behavior, organizational innovation climate, and self-transcendence among the Chinese clinical first-line nurses, (2) to examine the correlations among innovative behavior, organizational innovation climate, and self-transcendence, (3) to explore the mediating role of self-transcendence between innovative behavior and organizational innovation climate. In this study, we didn't compare the differences of innovative behavior of nurses with different demographic characteristics, and the results of univariate analysis of innovative behaviour are also not included in the subsequent section “5. Discussion”. Maybe this part is of little significance to the structure of the whole article, and the limitation of layout and number of tables, so the original section “4.3. Univariate analysis of innovative behaviour” is deleted and Table 1 is changed to table of d

---

## [Decision Letter · Decision Letter 2]

7 May 2024

PONE-D-23-28215R2Innovative behavior and organizational innovation climate among the Chinese clinical first-line nurses during the Omicron pandemic: The mediating roles of self-transcendencePLOS ONE

Dear Dr. Wang,

Thank you for submitting your manuscript to PLOS ONE. After careful consideration, we feel that it has merit but does not fully meet PLOS ONE’s publication criteria as it currently stands. Therefore, we invite you to submit a revised version of the manuscript that addresses the points raised during the review process.

We look forward to receiving your revised manuscript.

Kind regards,

Myriam M. Altamirano-Bustamante

Academic Editor

PLOS ONE

Reviewers' comments:

Reviewer's Responses to Questions

**Comments to the Author**

1. If the authors have adequately addressed your comments raised in a previous round of review and you feel that this manuscript is now acceptable for publication, you may indicate that here to bypass the “Comments to the Author” section, enter your conflict of interest statement in the “Confidential to Editor” section, and submit your "Accept" recommendation.

Reviewer #2: All comments have been addressed

2. Is the manuscript technically sound, and do the data support the conclusions?

Reviewer #2: Partly

3. Has the statistical analysis been performed appropriately and rigorously? 

Reviewer #2: Yes

4. Have the authors made all data underlying the findings in their manuscript fully available?

Reviewer #2: No

5. Is the manuscript presented in an intelligible fashion and written in standard English?

Reviewer #2: Yes

6. Review Comments to the Author

Reviewer #2: Major concerns

1. The introduction of the manuscript is verbose. The authors should consider the following suggestions:

-First, the background information provided is excessively detailed. The detailed descriptions of COVID-19, particularly the Omicron variant, are well-known and seem unnecessary in this manuscript. Since the intended audience of this study is likely already familiar with this information, it could be omitted. Instead, briefly state why the characteristics of COVID-19 and the Omicron variant are relevant in the context of this study.

-Second, there is repetitive information. The necessity for innovation in nursing and quality improvement is mentioned multiple times without each mention providing new information. While it is important to emphasize key points, repetition can scatter the reader's attention. Avoiding repetition and stating each point clearly once would enhance the clarity of the text.

-Lastly, unnecessary detailed descriptions should be avoided. Several cited studies are described in excessive detail. Unless these details are directly necessary for understanding this research, it would be more appropriate to succinctly state the main findings and how these studies contribute to the discussion of the paper. This approach would make the text more focused and easier to read.

2. There are some discrepancies between the statistical analysis design and the sample size calculations. In order to enhance the clarity, completeness, and scientific rigor of the study, it would be beneficial to consider the following points:

Analysis of Mediating Effects: The purpose of this study is not only to explore the correlation between innovative behavior, organizational innovation climate, and self-transcendence, but also to investigate the mediating role of self-transcendence between innovative behavior and organizational innovation climate. Given this objective, the sample size calculation should include considerations specific to mediation analysis. This type of analysis requires a sample size that adequately addresses the relationships among the independent, mediating, and dependent variables, taking into account the expected effect sizes of the paths involved. The description of sample size calculations in the current manuscript does not appear to meet these requirements. The following book is recommended as a reference: Hayes AF. Introduction to Mediation, Moderation, and Conditional Process Analysis: A Regression-Based Approach. 3rd ed. New York. : Guilford Publications; 2022 Jan 24. ISBN 9781462549030.

-Model Selection for Sample Size Calculation: The manuscript mentions that a "point biserial model" was used to calculate the sample size. However, this model is usually employed when one variable is binary and the other is continuous. In a mediation analysis, where usually all variables involved are continuous, the choice of this model may not be appropriate. A more detailed explanation of why this model was chosen and its suitability for the reported analysis is needed.

-Sample size discrepancy: the manuscript states that the "point biserial model" was used to calculate a minimum sample size of 266 participants, but the study included 1,058 participants. This large difference in numbers raises questions about the rationale for including such a large sample. While the large sample size may be justified by the requirement for a comprehensive mediation analysis, the manuscript currently lacks an explanation for this discrepancy. Clarifying why 1,058 participants were necessary would greatly enhance the transparency and understanding of the study design, especially in relation to the mediation analysis. understanding of the study design, especially in relation to the mediation analysis.

-Consistency of reporting: In my previous review I noted that the reporting of descriptive statistics should avoid mixing information from inferential statistics, and that this has been noted in STROBE as well. The authors accept this point and exclude the reporting of p-values from the summary of descriptive statistics results as seen in Table 1. If this is the case, reference to inferential statistical methods for between-group comparisons should also no longer be included, as it may confuse the reader or imply analytical methods not employed in the study.

Minor concerns

-"3.1. Research Design and Setting" should include detailed information about the setting. It is necessary to specify the time period (year and month) during which the study was conducted. Rather than merely stating 'China,' providing a more detailed location, such as the province or city, would be beneficial. Additionally, references to STROBE should be concise, only noting whether this report of the study is STROBE compliant.

-It is recommended to divide the section "3.2. Participants and Sample" into two different sections: “Participants” and “Sampling Methods and Sample Size Calculation.” In the “Participants” subsection, the inclusion and exclusion criteria should be detailed. In the “Sampling Methods and Sample Size Calculation” subsection, the focus should be on the procedures used to recruit participants and how the sample size was calculated. This restructuring will enhance the logical flow of the sections and make the information more organized and accessible.

-Although this is not a comment about the writing of the paper itself but rather about the authors' responses to my comments, I feel compelled to add one point. The authors frequently conclude their responses with the phrase, 'And our writing ability is limited, I hope you can be magnanimous.' While I understand this might stem from cultural differences, it is crucial to remember that the peer review process is designed to ensure the quality of research and provide objective feedback based on the content of the manuscript, not on the personal capabilities or feelings of the authors. I encourage the authors to focus on addressing the scientific feedback directly and ensure that communication remains professional and focused on the research itself. Such references to personal limitations in writing do not align with the professional standards expected in scholarly communication.

7. PLOS authors have the option to publish the peer review history of their article (what does this mean?). If published, this will include your full peer review and any attached files.

Reviewer #2: **Yes: **Chiharu Murata

---

## [Author Response · Author response to Decision Letter 2]

9 May 2024

Submission ID: PONE-D-23-28215 R3

Title: Innovative behavior and organizational innovation climate among the Chinese clinical first-line nurses during the Omicron pandemic: The mediating roles of self-transcendence 

May 9th, 2024

Dear academic editor and reviewers:

 We would like to thank you for your efforts in reviewing our revised manuscript entitled “Innovative behavior and organizational innovation climate among the Chinese clinical first-line nurses during the Omicron pandemic: The mediating roles of self-transcendence”, and providing many helpful comments and suggestions, which will all prove invaluable in the revision and improvement of our paper, as well as in guiding our research in the future.

According to your nice suggestions, we have studied your comments point-by-point, and have made extensive corrections to our previous draft, and all the changes have been marked in red in the revised draft, the detailed corrections are listed below. In order for reviewers and readers to better understand and read, we invited two nursing professors whose mother tongue is English and KetengEdit (www.ketengedit.com) to polish and revise the language of our article. All authors have approved the response letter and the revised version of the manuscript. 

Responds to the reviewers’ comments point-by-point:

Response to reviewer #2: 

 Dear reviewer, we appreciate it very much for these good suggestion. We are well aware that there are still some shortcomings in the manuscript. Therefore, in accordance with your suggestions, we have made the following revisions to our manuscript: 

Q1: “1. The introduction of the manuscript is verbose. The authors should consider the following suggestions:

-First, the background information provided is excessively detailed. The detailed descriptions of COVID-19, particularly the Omicron variant, are well-known and seem unnecessary in this manuscript. Since the intended audience of this study is likely already familiar with this information, it could be omitted. Instead, briefly state why the characteristics of COVID-19 and the Omicron variant are relevant in the context of this study.

-Second, there is repetitive information. The necessity for innovation in nursing and quality improvement is mentioned multiple times without each mention providing new information. While it is important to emphasize key points, repetition can scatter the reader's attention. Avoiding repetition and stating each point clearly once would enhance the clarity of the text.

-Lastly, unnecessary detailed descriptions should be avoided. Several cited studies are described in excessive detail. Unless these details are directly necessary for understanding this research, it would be more appropriate to succinctly state the main findings and how these studies contribute to the discussion of the paper. This approach would make the text more focused and easier to read.”

Response: Thank you very much for your valuable comments, for this suggestion, we have made major revisions, that is:

 (1) First, we have simplified the “Introduction” as a whole, from 1659 words to 1223 words.We have reorganized the section of “Introduction” and divided it into several paragraphs, which makes the overall structure more layered and logical. In order to make it easier to understand and read, we added the subtitle "1.1. Literature review" in the section of “Introduction”. We have also deleted the detailed descriptions of COVID-19, and stated why the characteristics of COVID-19 and the Omicron variant are relevant in the context of this study, that is, “Compared to Delta or SARS-CoV2, Omicron has more vital transmission ability, faster transmission speed, shorter incubation period, and rapid disease progression. These characteristics have had a profound negative influence on public health and have significantly impacted patients and clinical first-line nurses. By the end of 2022, many nurses in China were deployed to the frontlines of clinical care to assist in managing and containing the Omicron pandemic. However, clinical first-line nurses faced a greater risk of contracting Omicron than the general population. Not only do they have to endure the demanding workload brought about by the pandemic, but they also need to continuously innovate traditional nursing models and methods in clinical practice and enhance innovative awareness and ability to adapt to the pandemic-induced changes. And many studies have demonstrated that clinical first-line nurses were known to be at risk for depression, anxiety, fear, post-traumatic stress disorder, and enthusiasm for innovation to varying degrees during the Omicron pandemic, significantly higher than the SARS pandemic or general population. Studies have shown that, improving nurses' innovative behavior is a robust measure for addressing resource scarcity, heavy clinical nursing burden, and managing the Omicron pandemic challenges.”

 (2) Second, we have deleted repetitive information of necessity for innovation in nursing and quality improvement. In the first paragraph of the “Introduction”, the importance of nurses' innovative behavior is put forward. And put forward the limitations of existing research and the importance of this study, that is, “However, few research were on the innovative behavior of clinical first-line nurses and its related correlation study during the Omicron pandemic in China. Therefore, it is essential to investigate the innovative behavior of clinical first-line nurses and explore the relationships between other related variables.”

 (3) Lastly, we have succinctly state the main findings and how these studies contribute to the discussion of the paper. For several cited studies are described in excessive detail, we deleted some repetitive cited references, and the description was simplified. And the cited references in this study were reduced from 70 to 55, the order of all references has also been updated in the "References" section. In the third and fourth paragraphs of the “Introduction”, we put forward the general argument of the relationship between nurses' innovative behavior, organizational innovation climate and self-transcendence, and made it bold. In this way, we can clearly understand the research status and unresolved key issues of this study, which makes the overall structure more layered and logical.

 And all the changes have been marked in red in the revised draft.

Q2: “2. There are some discrepancies between the statistical analysis design and the sample size calculations. In order to enhance the clarity, completeness, and scientific rigor of the study, it would be beneficial to consider the following points:

Analysis of Mediating Effects: The purpose of this study is not only to explore the correlation between innovative behavior, organizational innovation climate, and self-transcendence, but also to investigate the mediating role of self-transcendence between innovative behavior and organizational innovation climate. Given this objective, the sample size calculation should include considerations specific to mediation analysis. This type of analysis requires a sample size that adequately addresses the relationships among the independent, mediating, and dependent variables, taking into account the expected effect sizes of the paths involved. The description of sample size calculations in the current manuscript does not appear to meet these requirements. The following book is recommended as a reference: Hayes AF. Introduction to Mediation, Moderation, and Conditional Process Analysis: A Regression-Based Approach. 3rd ed. New York. : Guilford Publications; 2022 Jan 24. ISBN 9781462549030.

-Model Selection for Sample Size Calculation: The manuscript mentions that a "point biserial model" was used to calculate the sample size. However, this model is usually employed when one variable is binary and the other is continuous. In a mediation analysis, where usually all variables involved are continuous, the choice of this model may not be appropriate. A more detailed explanation of why this model was chosen and its suitability for the reported analysis is needed.

-Sample size discrepancy: the manuscript states that the "point biserial model" was used to calculate a minimum sample size of 266 participants, but the study included 1,058 participants. This large difference in numbers raises questions about the rationale for including such a large sample. While the large sample size may be justified by the requirement for a comprehensive mediation analysis, the manuscript currently lacks an explanation for this discrepancy. Clarifying why 1,058 participants were necessary would greatly enhance the transparency and understanding of the study design, especially in relation to the mediation analysis. understanding of the study design, especially in relation to the mediation analysis.

-Consistency of reporting: In my previous review I noted that the reporting of descriptive statistics should avoid mixing information from inferential statistics, and that this has been noted in STROBE as well. The authors accept this point and exclude the reporting of p-values from the summary of descriptive statistics results as seen in Table 1. If this is the case, reference to inferential statistical methods for between-group comparisons should also no longer be included, as it may confuse the reader or imply analytical methods not employed in the study.”

Response: Thank you very much for your valuable comments, for this suggestion, we have made some major revisions, that is:

 (1) First, by studying this book “Hayes AF. Introduction to Mediation, Moderation, and Conditional Process Analysis: A Regression-Based Approach. 3rd ed. New York. : Guilford Publications; 2022”, we have changed “t-tests” to “F-tests” of G*Power 3.1.9.7 software, that is, “G*Power 3.1.9.7 software was used to calculate the minimum sample size required for F-tests [ α error probability was 0.05, and power (1 - β error probability) was 95% ]. Because all variables involved of this study are continuous, therefore, the F-tests method was selected in G*Power 3.1.9.7 software, “Linear multiple regression: Fixed model, R2 deviation from zero” of statistical test was selected, “A priori: Compute required sample size - given α, power, and effect size” of type of power analysis was selected, and then the parameter effect size f²= 0.15, α = 0.05, 1 - β = 0.95 were set, and finally clicked "Calculate" to calculate the total sample size = 107, the minimum sample size of this study.” 

 (2) Second, we clarified why 1,058 participants were necessary would greatly enhance the transparency and understanding of the study design, especially in relation to the mediation analysis, that is, “Considering the hospital and clinical realistic environment, 1,058 participants were included in this study in the end. And “Post hoc: Compute achieved power - given α, sample size, and effect size” of type of power analysis was selected. And then the parameter effect size f²= 0.15, α = 0.05, total sample size = 1058 were set, and finally clicked "Calculate" to calculate the 1 - β = 1.00, which was greater than the original 1 - β of 0.95 and indicating that the sample size of the study meets the sample size required for the mediation analysis. In addition, participants were recruited from seven tertiary grade-A hospitals of Tianjin city in Northern China through online questionnaires, and collected more questionnaires to ensure the generalization and universality of the research results, so as to minimize the deviation of the results caused by geography and hospital nature. In this way, included 1,058 participants were necessary and would greatly enhance the transparency and understanding of the study design, especially in relation to the mediation analysis. ”

 (3) Lastly, we have deleted inferential statistical methods for between-group comparisons in the section of “3.4. Statistical analysis”

 And all the changes have been marked in red in the revised draft.

Q3: “3.1. Research Design and Setting" should include detailed information about the setting. It is necessary to specify the time period (year and month) during which the study was conducted. Rather than merely stating 'China,' providing a more detailed location, such as the province or city, would be beneficial. Additionally, references to STROBE should be concise, only noting whether this report of the study is STROBE compliant.”

Response: Thank you very much for your valuable comments, for this suggestion, we have made some revisions, that is, “ This study used a descriptive cross-sectional design and was conducted of Tianjin city in Northern China from March 2022 to February 2023. The quality reporting of the study adhered to the STROBE Checklist (see S1).” 

And all the changes have been marked in red in the revised draft.

Q4: “It is recommended to divide the section "3.2. Participants and Sample" into two different sections: “Participants” and “Sampling Methods and Sample Size Calculation.” In the “Participants” subsection, the inclusion and exclusion criteria should be detailed. In the “Sampling Methods and Sample Size Calculation” subsection, the focus should be on the procedures used to recruit participants and how the sample size was calculated. This restructuring will enhance the logical flow of the sections and make the information more organized and accessible.”

Response: Thank you very much for your valuable comments, for this suggestion, we have made some revisions, that is: 

 “3.2. Participants and sample

 3.2.1. Participants 

 Convenience sampling was used to recruit the nurses from seven tertiary grade-A hospitals during the Omicron pandemic, which are all located in Tianjin city of Northern China, with more than 1,500 beds and 2,000 employees. Respondents met the following criteria—inclusion criteria: (1) obtained a nurse's professional qualification and register certificates; (2) Worked in the first-line during the Omicron pandemic for ≥ 1 month. Exclusion criteria: (1) intern nurses, (2) advanced training, rotation, and regular training nurses, and (3) nurses who were not on duty during the investigation period. Written informed consents were taken from the nurses who volunteered to participate in the study.

3.2.2. Sampling methods and sample size calculation

Participants were recruited from seven tertiary grade-A hospitals of Tianjin city in Northern China from March 2022 to February 2023. The investigation was conducted with the hospitals' prior approval, focusing on individual departments and with the assistance of the head nurse in each department for distribution. We used the unified instruction language to explain the essential information to the participants, including the purpose, significance, and confidentiality of this study, and the participants were investigated face-to-face using an online questionnaire. The questionnaires underwent quality control through several measures: (1) each micro-signal can be filled only once, after being authorized by the system settings; (2) the participants are required to answer all the questions; (3) questionnaires with completion time less than one minute were excluded; (4) questionnaires with obvious contradictory answers were excluded; (5) questionnaires with obvious inconsistent personal information of the respondents were excluded. Finally, 1,058 valid questionnaires were collected, and the effective recovery rate was 96.7%.

 G*Power 3.1.9.7 software was used to calculate the minimum sample size required for F-tests [ α error probability was 0.05, and power (1 - β error probability) was 95% ]. Because all variables involved of this study are continuous, therefore, the F-tests method was selected in G*Power 3.1.9.7 software, “Linear multiple regression: Fixed model, R2 deviation from zero” of statistical test was selected, “A priori: Compute required sample size - given α, power, and effect size” of type of power analysis was selected, and then the parameter effect size f²= 0.15, α = 0.05, 1 - β = 0.95 were set, and finally clicked "Calculate" to calculate the total sample size = 107, the minimum sample size of this study. 

Considering the hospital and clinical realistic environme

---

## [Decision Letter · Decision Letter 3]

12 Jun 2024

Innovative behavior and organizational innovation climate among the Chinese clinical first-line nurses during the Omicron pandemic: The mediating roles of self-transcendence

PONE-D-23-28215R3

Dear Dr. Wang,

We’re pleased to inform you that your manuscript has been judged scientifically suitable for publication and will be formally accepted for publication once it meets all outstanding technical requirements.

Kind regards,

Myriam M. Altamirano-Bustamante

Academic Editor

PLOS ONE

Additional Editor Comments (optional):

Reviewers' comments:

Reviewer's Responses to Questions

**Comments to the Author**

1. If the authors have adequately addressed your comments raised in a previous round of review and you feel that this manuscript is now acceptable for publication, you may indicate that here to bypass the “Comments to the Author” section, enter your conflict of interest statement in the “Confidential to Editor” section, and submit your "Accept" recommendation.

Reviewer #2: All comments have been addressed

2. Is the manuscript technically sound, and do the data support the conclusions?

Reviewer #2: Yes

3. Has the statistical analysis been performed appropriately and rigorously? 

Reviewer #2: Yes

4. Have the authors made all data underlying the findings in their manuscript fully available?

Reviewer #2: Yes

5. Is the manuscript presented in an intelligible fashion and written in standard English?

Reviewer #2: Yes

6. Review Comments to the Author

Reviewer #2: I have made high-level demands of the authors during this peer review process, and you have responded well to those demands. In my judgment, you have submitted manuscripts that are suitable for publication in PLOS ONE. I would like to endorse PLOS ONE's acceptance of this version.

7. PLOS authors have the option to publish the peer review history of their article (what does this mean?). If published, this will include your full peer review and any attached files.

Reviewer #2: **Yes: **Chiharu Murata

---

## [Editor Report · Acceptance letter]

21 Jun 2024

PONE-D-23-28215R3 

PLOS ONE

Dear Dr. Wang, 

I'm pleased to inform you that your manuscript has been deemed suitable for publication in PLOS ONE. Congratulations! Your manuscript is now being handed over to our production team.

Kind regards, 

on behalf of

Dr. Myriam M. Altamirano-Bustamante 

Academic Editor

PLOS ONE